# Anomaly-focused Single Image Super-resolution with Artifact Removal for Chest X-rays using Distribution-aware Diffusion Model

**Dattatreyo Roy**                                                   m22cs060@iitj.ac.in
**Angshuman Paul**                                                   apaul@iitj.ac.in
*Indian Institute of Technology Jodhpur, Jodhpur, India*

**Editors:** Accepted for publication at MIDL 2024

## Abstract

Single image super-resolution (SISR) is a crucial task in the field of medical imaging. It transforms low-resolution images into high-resolution counterparts. Performing SISR on chest x-ray images enhances image quality, aiding better diagnosis. However, artifacts may be present in the images. We propose an anomaly-guided SISR process utilizing the denoising mechanism of the diffusion model to iteratively remove noise and restore the original image. We train the model to learn the data distribution, enabling it to eliminate artifacts within the images. Additionally, we ensure reconstruction of the disease regions by prioritizing their reconstruction. Our research experiment over the publicly available dataset and find that the existing SISR methods are unable to learn and remove these artificially added artifacts. On the other hand, our proposed model not only prioritizes superior image reconstruction but also remove the artifacts. Our method is found to outperform the existing methods. The code is publicly available at https://github.com/Datta-IITJ/MIDL_code.git.

**Keywords:** Diffusion Model, variational autoencoding, artifact removal, bounding box loss, chest x-ray

## 1. Introduction

Medical images with superior resolution may provide important information about various abnormalities that may be present in such images. Such information is likely to play a crucial role in the diagnosis. Chest x-ray is one of the most widely used imaging modalities. Chest x-rays with a superior resolution may facilitate the diagnosis of various abnormalities by radiologists. Furthermore, in various applications including telemedicine, it may be required to compress the size of such images. However, at the time of diagnosis, the original resolution of those images should be restored. Therefore, improving the resolution of chest x-ray images may potentially aid various aspects of modern healthcare.

Single image super-resolution (SISR) methods aim to create high resolution (HR) images from their low resolution (LR) counterparts. In spite of the significant progress in the field of SISR (Ledig et al., 2017; Wang et al., 2018; Saharia et al., 2022; Li et al., 2022), such methods are relatively rare for medical images including chest x-rays. In (Yu et al., 2021), the authors presented WFSAN, a lightweight architecture for high-quality medical image super-resolution. SNSR-GAN was proposed in (Xu et al., 2020) for enhancing chest x-ray images. The authors of (Monday et al., 2022) introduced the COVID-SRWCNN which employs a siamese wavelet multi-resolution convolutional neural network. For anomaly-driven SISR of chest x-ray images, see (Yadagiri et al., 2023).

Due to various factors including image compression, and presence of foreign objects, artifacts may be created in chest x-rays and other medical images. However, most of the existing methods do not involve a mechanism to deal with artifacts in LR images during the process of generating the super-resolved (SR) images. As a result, such artifacts may be present in SR images. The presence of such artifacts may affect the diagnosis of the medical images.

We propose a SISR method for chest x-rays. A major goal in our design is to remove artifacts during the super-resolution process. Diffusion models (Ho et al., 2020) take a noisy image and learn to iteratively denoise it. Therefore, such models may be useful for removing artifacts by treating the artifacts as a form of noise. Hence, we design our SISR method utilizing diffusion models. We propose a novel training strategy to deal with the artifacts.

While designing the proposed model, we consider the fact that learning the distribution of data may aid the super-resolution process. We also note that emphasizing the abnormality during the super-resolution process may result in an SR image that has rich information about the abnormality. Such an SR image may lead to an improved diagnosis. Most diffusion models are designed using a U-net backbone. We modify the U-net backbone with variational autoencoding (Kingma and Welling, 2013) mechanism to capture the distribution of the data. We also design a loss function that helps to focus on the region with abnormality during the super-resolution process. This may lead to an SR image with richer information about the abnormalities. In this work, our major contributions are:

- We introduce a SISR method for chest x-rays using a diffusion probabilistic model that can remove artifacts during super-resolution.

- The proposed model utilizes information about abnormalities that may be present in the chest x-rays. The resultant SR image is likely to contain richer information about the abnormalities.

- Our model captures the distribution of data using a variational autoencoding mechanism to facilitate super-resolution.

- Experiments on publicly available datasets show the potential of generalizability of the proposed method.

The rest of the paper is organized as per sections. Section 2 consists of the methodology where we mention details regarding our proposed method and the training process. Section 3 is the experimental details and the quantitative and qualitative results obtained using our methodology. Section 4 contains the conclusion.

## 2. Methods

We design a method for the single image super-resolution of chest x-rays. To design this method, we consider the following facts. Due to various reasons, artifacts may be present in LR chest x-ray images. Removal of these artifacts during super-resolution may facilitate the diagnosis. Diffusion models can iteratively remove noise from images. Therefore, diffusion models may be helpful in removing artifacts. So, we use a diffusion model-based approach for SISR. We also note that in the super-resolved images, if we have rich information about

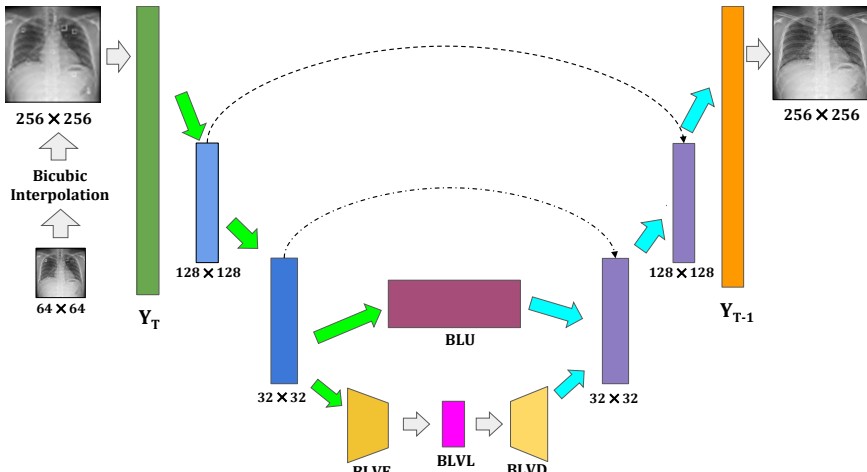

Figure 1: Block diagram of the proposed backbone model for one iteration of the diffusion model. Input images of size 64×64 pixels are interpolated to 256×256 pixels before being fed into the model. The model comprises a modified U-net architecture with additional layers for encoding (BLVE), latent space (BLVL), and decoding (BLVD) layers to incorporate variational autoencoding. The original bottleneck layer of the U-net backbone is denoted as BLU. During the reverse process, at each iteration, the model is expected to generate a less noisy image $Y_{T-1}$ from a more noisy image $Y_T$ obtained from the previous iteration. The eventual output after all the iterations of the reverse process is the SR image. The Green and the blue arrows indicate only the down-sampling and up-sampling operation, respectively. The Dashed lines in black denote the skip-connections.

the abnormality present in the x-ray images, diagnosis may be improved. Therefore, we design an anomaly-guided SISR method. Furthermore, we also consider that capturing the distribution of the data during the super-resolution process may help in achieving a superior SR image. Thus, as a backbone of our model, we utilize a U-net based architecture (Ronneberger et al., 2015) that explicitly capture the distribution of the data. A block diagram of this architecture is presented in Fig.1.

## 2.1. Diffusion Probabilistic Model

We build our super-resolution model on the design of (Saharia et al., 2022). During training, we employ a forward Markovian diffusion process to gradually add Gaussian noise to a high-resolution (HR) chest x-ray image over $T$ iterations. In the reverse process, we iteratively denoise the above noisy image to get back the HR image through $T$ iterations following (Saharia et al., 2022). At each iteration of the reverse process, the model is expected to generate a less noisy image $Y_{T-1}$ from a more noisy image $Y_T$ obtained from the previous iteration. The eventual output (after all the iterations of the reverse process) of the model is the Super-resolved image. During the inference process, we take an LR chest x-ray image

and perform bicubic interpolation to improve the resolution. The bicubic interpolation only provides an approximate super-resolved image. After adding noise, this interpolated image becomes noisy. Subsequently, the reverse process is applied to this noisy image to generate a super-resolution (SR) image from it. The method of (Saharia et al., 2022) is designed using a U-net backbone to perform the reverse process. We modify the design of (Saharia et al., 2022) in such a way that we capture the distribution of the data during the process of super-resolution. We also emphasize the abnormal regions during this process.

## 2.2. Capturing Data Distribution

To capture the distribution of the data through our U-net backbone, we employ a variational autoencoding mechanism. Variational autoencoders (VAE) (Kingma and Welling, 2013) can capture the distribution of data. We modify the structure of the bottleneck layer of our U-net backbone to act like the latent layer of VAE. In this context, we note that U-net consists of an encoder-decoder architecture. Therefore, if we can enforce a distributional similarity in the bottleneck layer of U-net, the modified U-net can emulate the properties of VAE in the context of capturing data distribution. Therefore, in our method, the latent layer of the VAE captures the distribution of the input data after the data is transformed by the encoder part of U-Net.

To that end, we add a few layers parallel to the bottleneck layer of the U-net (see Fig.1). We abbreviate the original bottleneck layer of U-net as BLU and the newly added layers as BLVE, BLVL and BLVD. Layers BLVE and BLVD act like an encoder and a decoder layer, respectively. Layer BLVL acts like the latent layer of a VAE. Let $Z$ be the representation of transformed input data at BLVL. We calculate the KL divergence loss ($L_{KL}$) between $Z$ and a standard normal reference distribution. Minimization of this makes the distribution of $Z$ similar to the reference distribution. Loss $L_{KL}$ helps in capturing the distribution of this data in latent layer BLVL. Since the output of BLVD is generated by sampling from the distribution learnt in BLVL, the output quality from BLVL and thereby the output of the proposed method is likely to be dependent on the distribution of the data captured in BLVL. To enforce autoencoding, we also calculate a reconstruction loss ($L_{VR}$) between the input to BLVE and the output from BLVD.

## 2.3. Anomaly-focused Training

A better reconstruction of the regions with abnormality may aid the diagnosis. So, we aim to provide additional emphasis on the reconstruction of such regions. To that end, we utilize the bounding box (BB) annotations of the abnormalities in the chest x-ray images. We calculate the mean-squared error of the pixel intensities inside BB between the SR and the corresponding HR images. This error serves as bounding box loss ($L_B$). We ignore the BB-loss component if an image does not have the BB information (e.g., images with no anomaly).

In addition, we also compute a reconstruction loss ($L_R$) between the original HR image without artifacts and the SR image generated by our method. Minimization of this loss helps in making the SR image similar to the HR image. During the training of the proposed method, we minimize the following loss:

$$L_{\text{total}} = \lambda_R L_R + \lambda_B L_B + \lambda_{VAE}(L_{VR} + L_{KL}), \tag{1}$$

where $\lambda_R$, $\lambda_B$, and $\lambda_{\mathrm{VAE}}$ represent the weights governing the relative importance of each component. $L_R$ and $L_B$ are already mentioned above. $L_{\mathrm{VR}}$ is the reconstruction loss between the representations of the input data created by layers BLVE and BLVD. $L_{\mathrm{KL}}$ is the KL divergence loss between the representation $Z$ at BLVL and a standard normal reference distribution.

## 2.4. Artifact Removal

A major goal of the proposed method is to remove artifacts during super-resolution. For that purpose, we propose the following training protocol. First, we add artifacts in the LR images. The original chest x-ray artifacts have circular shapes, letters, digits, and lines (art; bra). We tried to create similar artifacts. During the training, we use these images with artifacts as input and try to generate super-resolved images without artifacts. Our diffusion model, being suitable for removing noise is expected to treat the artifacts as noise and learn to remove those artifacts during the process of super-resolution.

## 2.5. Inference

During inference, we apply an LR chest x-ray image with artifacts to our trained model. First, a bicubic interpolation is performed to create an approximated super-resolved image. Noise is added to this image. We continue adding the noise until we create a fully noisy image up to T time-steps. Subsequently, the SR image is generated from the interpolated noisy image through the reverse process. A step-wise visualization of the diffusion process is presented in Appendix A.

## 2.6. Implementation Details

The backbone of the proposed model is a U-net structure similar to the one used in (Saharia et al., 2022). However, from the down-sampled stage, we create a branch for implementing variational autoencoding. This branch has got one encoding convolutional layer BLVE, one decoding convolutional layer BLVD, and a fully connected latent layer BLVL. We then concatenate the reconstructed outputs from the bottleneck layer of the U-net and BLVD layer. It then goes to the up-sampling stage where it reconstructs the image back to the required dimension.

To add noise to the images, we use a linear schedule adopted over 1000 time-steps denoted as T in Fig.1. The noise level initiates at $1 \times 10^{-4}$ and gradually increases to $1 \times 10^{-2}$. Our model is trained using the Adam optimizer with a learning rate of 1e-4 and batch size of 2. We include the hyperparameter details of the competitive methods in the Appendix B. For the calculation of loss, we use $\lambda_R$ as 0.4, $\lambda_B$ as 0.3, and $\lambda_{VAE}$ as 0.3. All these parameters are selected based on the validation performance.

## 3. Experiment and Results

### 3.1. Datasets

We use two publicly available chest x-ray datasets for our experiments. These are VinBig chest x-ray dataset (Nguyen et al., 2022) and NIH Chest x-ray14 dataset (Wang et al.,

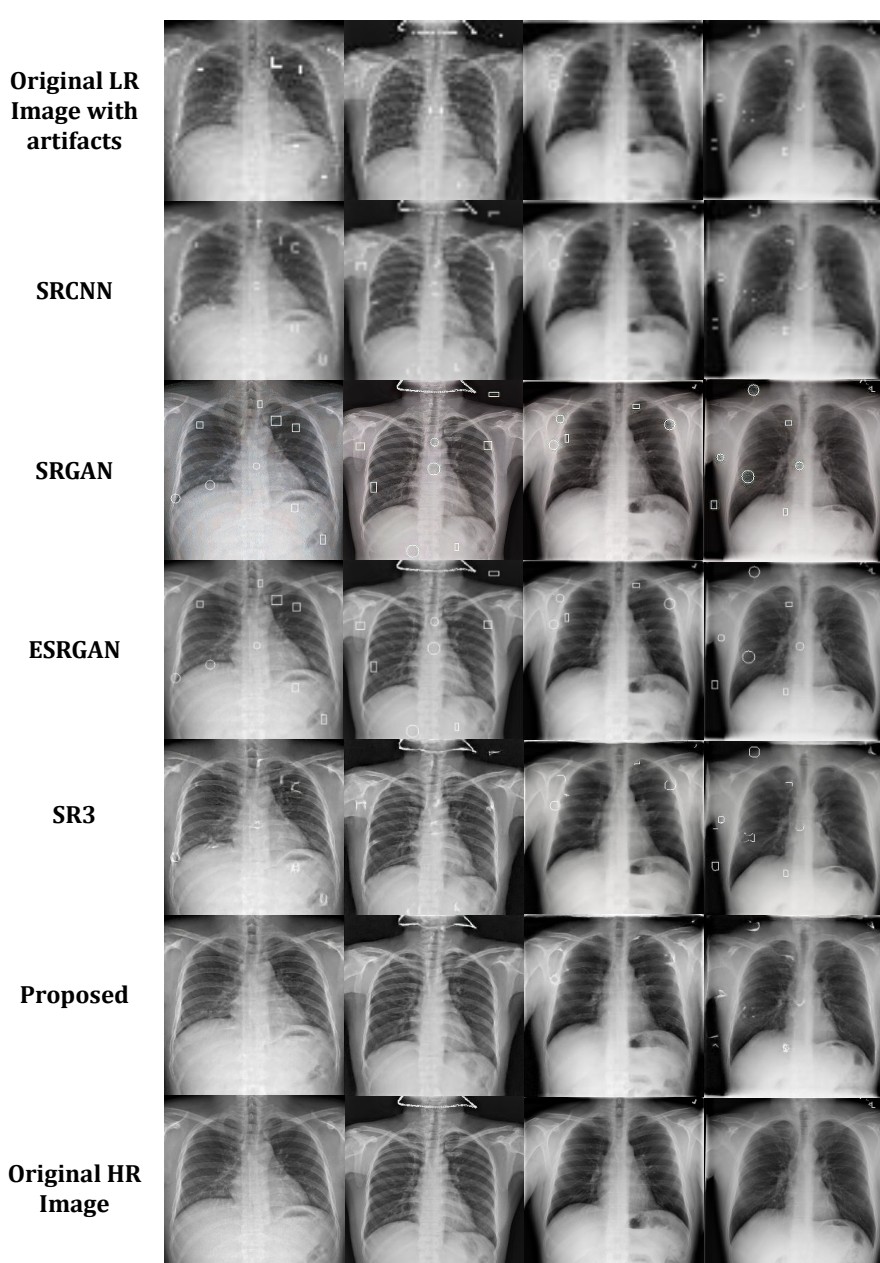

Figure 2: Result of SISR from 64×64 to 256×256 resolution on the test set from VinBig (columns 1-2) and NIH (columns 3-4) datasets using various methods. Original LR images with artifacts have a size of 64×64. The HR images and the super-resolved images using various methods (rows 3-7) have a dimension of $256 \times 256$.

2017). The VinBig dataset consists of 18,000 postero-anterior (PA) chest X-ray images in DICOM format, categorized into 15 classes representing various medical conditions. The dataset contains 18,000 images. For our experiments, we use 12,000 training images, 3,000 validation images, and 3,000 test images. Our model is trained with the training images of

Table 1: Performances of different models in terms of PSNR and SSIM (mean ± sd) computed between the ground truth HR image of size 256×256 without artifacts and the SR output using the corresponding models. The results are reported using the test set from VinBig and NIH datasets. **All the values in this table are computed on the test data of VinBig and NIH Dataset using the model trained on the VinBig dataset.**

| | VinBig | | NIH | |
|---|---|---|---|---|
| Model | PSNR | SSIM | PSNR | SSIM |
| Bicubic | $22.684 \pm 0.553$ | $0.631 \pm 0.009$ | $21.741 \pm 0.443$ | $0.626 \pm 0.006$ |
| SRCNN | $25.057 \pm 0.493$ | $0.655 \pm 0.002$ | $24.144 \pm 1.024$ | $0.656 \pm 0.003$ |
| SRGAN | $31.813 \pm 1.116$ | $0.719 \pm 0.001$ | $30.104 \pm 0.541$ | $0.713 \pm 0.009$ |
| ESRGAN | $33.688 \pm 1.119$ | $0.737 \pm 0.001$ | $32.679 \pm 1.107$ | $0.721 \pm 0.002$ |
| SR3 | $37.717 \pm 0.584$ | $0.797 \pm 0.005$ | $35.405 \pm 0.829$ | $0.785 \pm 0.008$ |
| **Proposed** | $\mathbf{38.936 \pm 0.914}$ | $\mathbf{0.813 \pm 0.002}$ | $\mathbf{36.532 \pm 0.789}$ | $\mathbf{0.805 \pm 0.007}$ |

the VinBig dataset. The NIH dataset comprises 112,120 x-ray images with disease labels from 30,805 patients. We use 3000 images from the NIH dataset for testing only.

### 3.2. Comparative Performances

We compare the performance of the proposed method with several state-of-the-art SISR techniques including SRCNN (Dong et al., 2015), SRGAN (Ledig et al., 2017), ESR-GAN (Wang et al., 2018) and SR3 (Saharia et al., 2022). The details of hyperparameters for the competing methods are presented in Appendix C. The performances are evaluated based on Structural Similarity Index (SSIM) and Peak Signal-to-Noise Ratio (PSNR) calculated using SR image produced by a method and the ground truth HR image (without artifacts). All the models are trained on the VinBig training dataset and tested on the Vin-Big test dataset and the NIH test dataset. We train the proposed method and all SOTA methods using the same training images with same artifacts. In all the experiments, an LR input image of size 64×64 is created from the original HR image by down-sampling. Subsequently, artifacts are added to these LR images. We train the various models to create super-resolved images of size 256×256 from these LR images. Thus, the numerical and visual results not only show the efficacy of the proposed method for super-resolution but also show its effectiveness in removing artifacts while performing super-resolution.

The results using the different methods for ten runs are reported in Table 1 in terms of PSNR and SSIM. Notice that the proposed method outperforms all its competitors on both the datasets. Results on sample images using different methods are presented in Fig.2. Additional results showing the various images with and without artifacts and the output of the proposed method are presented in Fig.4 of Appendix C. A statistical analysis between the performance of SR3 and the Proposed method is presented in Appendix E.

### 3.3. On Generalizability

Since we train the SISR models with the VinBig dataset, the results on the NIH dataset is an indicator of the generalizability of the SISR methods. The abnormalities present in

Table 2: Performances in different ablation studies in terms of PSNR and SSIM (mean ± sd) for the VinBig dataset.

| Model | PSNR | SSIM |
|---|---|---|
| SR3 | 37.717 ± 0.584 | 0.797 ± 0.005 |
| W-BBLoss | 38.741 ± 0.862 | 0.807 ± 0.007 |
| W-DataDist | 37.954 ± 0.237 | 0.793 ± 0.009 |
| Proposed | 38.936 ± 0.914 | 0.813 ± 0.002 |

the VinBig and NIH Dataset are mentioned in Appendix D. We note that there are some unseen anomalies which are not present in the VinBig dataset but are present in the NIH dataset (e.g, edema, hernia). So, when we use our model trained on the VinBig dataset and perform testing on the NIH dataset, we encounter these unseen abnormalities in the test data. From Table 1, notice that the proposed method outperforms all the competing approaches in this context. Thus, we conclude that the test results on the NIH dataset show the ability of our model to generalize to unseen anomalies as well.

### 3.4. Ablation Studies

We perform various ablation studies to look into the importance of different components of the proposed method. All the ablation studies are performed on the VinBig dataset. First, we evaluate the importance of the bounding box loss of (1). To that end, we train our model excluding the bounding box loss (abbreviated as W-BBLoss). We also look into the impact of capturing the data distribution in our model. For this purpose, we train our model without the variational autoencoding branch (abbreviated as W-DataDist). The results of these ablation studies are presented in Table 2. Notice that for both of the ablation studies, we obtain inferior results compared to the proposed method. These results signify the importance of different components of the proposed method.

### 4. Conclusion

We introduce a SISR method for chest x-rays that eliminates artifacts during super-resolution. Our method employs diffusion model to facilitate an iterative denoising. A novel bounding box helps to emphasize the abnormal regions and produce richer information about the abnormalities in the SR images. We design a variational autoencoding mechanism in our architecture to capture the underlying data distribution during super-resolution. A novel training strategy helps in removing the artifacts. Rigorous experiments show not only the usefulness of our method in publicly available datasets but also its generalizability. Ablation studies show the impact of different components in our design. In the future, we will explore the possibility of using similar methods in other radiology images including CT. We will also look into the possibility of utilizing auxiliary information for SISR.

## Acknowledgments

We thank the National Institutes of Health Clinical Center for providing the NIH dataset. This work is supported by SEED grant from IIT Jodhpur.

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

## Appendix A. Step-wise Visualization of the Diffusion Process

Fig.3 shows the forward and reverse process in a diffusion model. In the Forward Diffusion Process, the model orchestrates the gradual evolution of a random initial state towards a desired target state. Controlled amounts of Gaussian noise are added at each step, following a Markov chain. In contrast, during inference, the reverse diffusion process is employed to recover the original target state from a noisy observation. The model iteratively refines the noisy image by reversing the introduced noise using a denoising model. From Fig. 3, it can be observed that during noise removal, our method removes the artifacts also.

## Appendix B. Hyperparameters for the Competing Methods

The SRCNN model is trained over 450 epochs, with a learning rate of 0.001. The batch size is set to 4 and the Adam optimizer is employed to minimize Mean Squared Error (MSE) loss. Unlike SRCNN, SRGAN is trained for a shorter duration of 100 epochs. A lower learning rate of 0.0002 is utilized, with the batch size maintained at 4. Similar to SRCNN, the Adam optimizer is used. Training of ESRGAN is continued for 120 epochs with a learning rate of 0.005. A batch size of 8 is used with the Adam optimizer. The SR3 model is trained for 250 epochs. We employ a learning rate of 1e-5. Unlike the previous methods, SR3 utilizes the L1 loss function. The batch size is set to 1, and the Adam optimizer is used.

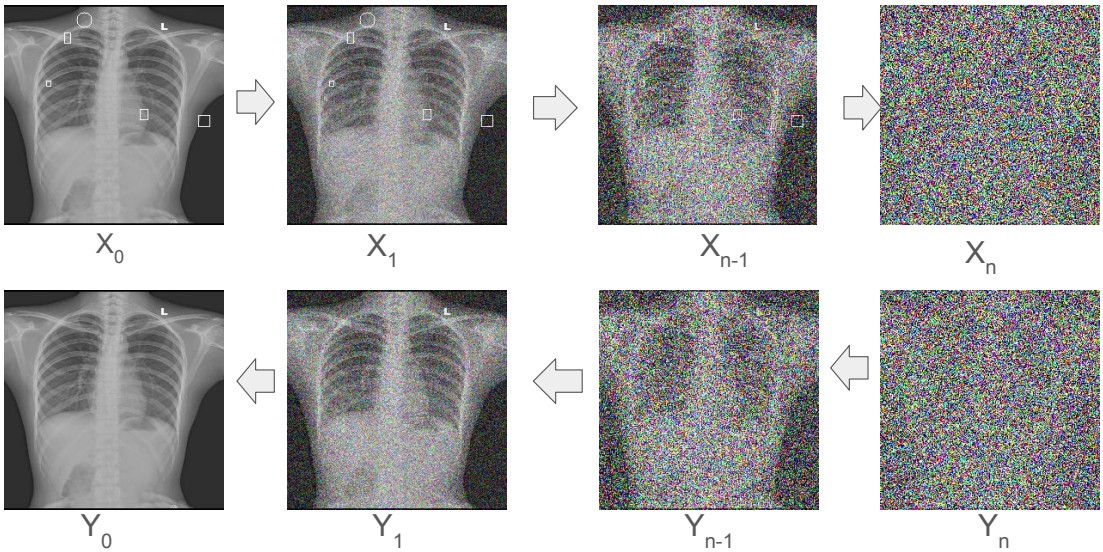

Figure 3: Diagram of the diffusion process. Top row: forward process with images at multiple time-steps, bottom row: reverse process with images at multiple time-steps.

## Appendix C. Result on Sample Images using the Proposed method

Fig.4 shows the results of the proposed method on the test set of the VinBig dataset.

## Appendix D. Labels Present in VinBig and NIH dataset

The VinBig Dataset consists of the following abnormalities: Aortic enlargement, Atelectasis, Calcification, Cardiomegaly, Consolidation, ILD, Infiltration, Lung Opacity, Nodule/Mass, Other lesion, Pleural effusion, Pleural thickening, Pneumothorax and Pulmonary fibrosis. The NIH dataset consists of the following abnormalities: Atelectasis, Consolidation, Infiltration, Pneumothorax, Edema, Emphysema, Fibrosis, Effusion, Pneumonia, Pleural thickening, Cardiomegaly, Nodule Mass and Hernia.

## Appendix E. Statistical Analysis of the Comparative Performances between SR3 and the Proposed method.

We also investigate if the performance of the proposed method is statistically significantly different compared to that of our baseline SR3 method. As mentioned in Section 3.2, for each method, we perform ten rounds of experiments. At each round, we evaluate the performance on the VinBig and NIH datasets through PSNR and SSIM. Using those values of PSNR and SSIM values for both the VinBig and NIH test data, we perform a t-test to look into the statistical significance of the difference in performance between our method and SR3. The p-values of these experiments are reported in Fig.5. For the NIH dataset, the p-values for PSNR and SSIM are 0.0029 and 0.0018, respectively. Similarly, for the VinBig

Figure 4: Result of SISR from 64×64 to 256×256 resolution using the proposed method on the test data from VinBig dataset showing images at various stages. Original LR and original LR with artifacts have a size of 64×64. Original HR, Original HR with artifacts, and Proposed have a size of 256×256.

dataset, the p-values for PSNR and SSIM are 0.0011 and 0.0158, respectively. Therefore, we can conclude that the results of our method are statistically significantly different from those of SR3.

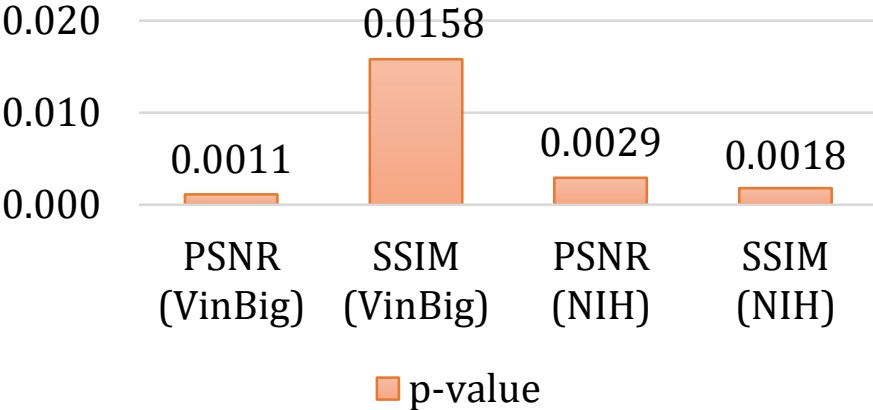

Figure 5: T-test between the SR3 baseline and Proposed method using the PSNR and SSIM values for both the VinBig and NIH test data.

