# OpenReview forum: "Anomaly-focused Single Image Super-resolution with Artifact Removal for Chest X-rays using Distribution-aware Diffusion Model"
_MIDL.io/2024/Conference — MIDL 2024 Poster_

### Official Review · Reviewer_FwCt · 2024-02-28

**Confidence:** 5
**Preliminary Rating:** 3
**Final Rating:** 4

**Summary:**

This paper proposes a modified diffusion model for Chest X-ray super-resolution, incorporating artifact augmentation, a variational architecture as the diffusion backbone, and an anomaly-focused loss. Both an ablation study and in-domain and out-of-domain test comparisons were provided, demonstrating better performance of the proposed method compared to other method.

**Strengths:**

- The paper compared the proposed method with several benchmark approaches and demonstrates its superior performance in both in-domain and out-of-domain scenarios.
- An ablation study was conducted to demonstrate the effectiveness of the proposed building blocks.

**Weaknesses:**

- U-Net architecture: In Figure 1, it appears that the U-Net architecture lacks skip connections from the encoder to the decoder. If skip connections were not used, the architecture should be referred to as an autoencoder instead of a U-Net.
- The empirical analysis that VAE captures data distribution is misleading. The authors used VAE on the bottleneck features instead of on the original image data, so the VAE used here should work as a regularization on the latent space feature, not necessarily capturing the `data distribution’.
- Training other SOTA methods: Did you include anomaly augmentation and bounding box loss when training other SOTA methods too?
- Generalization to unseen anomaly: It is not clear whether the trained model can generalize to unseen anomaly.

**Detailed Comments:**

See comments above

**Justification Of Final Rating:**

I thank the authors for their response and additional experiments on generalization test during this short rebuttal period. I'm satisfied with the new results/descriptions and would like to increase my rating to '4: weak acceptance'.

**Justification Of The Preliminary Rating:**

It is not clear whether the authors made a fair comparison to other SOTA methods by using the same augmentation and loss. Additionally, clarity is needed regarding the architecture used and the empirical analysis of latent space VAE.

**Questions To Address In The Rebuttal:**

Please respond to the comments regarding skip connections and training other SOTA methods. Please also consider modifying the empirical analysis of using latent space VAE. Future work should include testing generalization to unseen artifacts.

---

> ### Author Response · Authors · 2024-03-16
> **Addressing the comments mentioned in the "weaknesses" section.**
>
> $\textbf{1. Comments of the reviewer:}$ U-Net architecture: In Figure 1, it appears that the U-Net architecture lacks skip connections from the encoder to the decoder. If skip connections were not used, the architecture should be referred to as an autoencoder instead of a U-Net.
>
> $\textbf{Author's response:}$
> We are sorry for the confusion. Our design is motivated from the SR3 model [1] which uses U-Net as backbone. We also use U-Net architecture having skip connections. We have now modified Fig. 1 to show the skip connections.
>
> $\textbf{2. Comments of the reviewer:}$
> The empirical analysis that VAE captures data distribution is misleading. The authors used VAE on the bottleneck features instead of on the original image data, so the VAE used here should work as a regularization on the latent space feature, not necessarily capturing the `data distribution’.
>
> $\textbf{Author's response:}$ We agree that we use the VAE on the bottleneck features. However, the bottleneck features are generated through transformation of the input data by the neural network layers of the encoder of U-Net.  This is very similar to what happens in a vanilla VAE [2]. Therefore, in our method, the latent layer of the VAE captures the distribution of the input data after the data is transformed by the encoder part of U-Net. By ‘data distribution’, we mean the distribution of this transformed input. We will add this detail in Section 2.2 of the revised manuscript. We thank the reviewer for the comment.
>
>
> $\textbf{3.Comments of the reviewer:}$ Training other SOTA methods: Did you include anomaly augmentation and bounding box loss when training other SOTA methods too?
>
> $\textbf{Author's response:}$ We have trained the proposed method and all SOTA methods using the same dataset (same set of training images) with the same anomaly augmentations.  We revise section 3.2 to add these details.
>
> On the other hand, the use of bounding box loss is a contribution of the proposed method and the vanilla SOTA methods do not include such a loss. Therefore, while doing the comparisons, we do not include bounding box loss in SOTA methods.
>
> $\textbf{4.Comments of the reviewer:}$ Generalization to unseen anomaly: It is not clear whether the trained model can generalize to unseen anomaly.
>
> $\textbf{Author's response:}$ We thank the reviewer for pointing this out. In order to look into the generalizability of the proposed method, we perform experiments by training our model with the VinBig dataset and testing the trained model on the NIH dataset. Note that there are some abnormalities which are not present in the VinBig dataset but are present in the NIH dataset (e.g, edema, hernia). Therefore, when we perform testing on the NIH dataset (with the model trained in VinBig dataset), we encounter some unseen anomalies (such as edema, hernia) in the test data. Thus, the test result on the NIH dataset shows the ability of different methods (see Table 1) including our method to generalize to unseen anomalies as well. From Table 1, notice that our method outperforms all the competing approaches in this context. We revise section 3.3 to include this discussion.
>
> In our future work, we will test generalization on unseen artifacts. We thank the reviewer for the suggestion.
>
> $\textbf{Reference:}$
> 1. https://arxiv.org/abs/2104.07636
>
> 2. https://arxiv.org/abs/1312.6114

---

### Official Review · Reviewer_wbCG · 2024-02-29

**Confidence:** 4
**Preliminary Rating:** 4
**Recommendation:** Poster
**Final Rating:** 4

**Summary:**

This paper proposes using an image denoising diffusion model to perform image super-resolution and remove artifacts at the same time.

The network architecture is a U-net with an additional VAE architecture in the latent space.

The model is trained with pairs of (artifact-added low-resolution image, clean high resolution image).

**Strengths:**

Thorough comparisons: The model is compared against 5 previous super resolution models. Ablation studies were performed for different components of the model. Two chest x-ray datasets are used - VinBig to train, and VinBig/NIH to test.

Performance improvements are clear based on the table and figures. Ablation studies indicate that all parts contributed to the increased performance.

The main ideas and results are communicated clearly.

**Weaknesses:**

It's unclear how close the generated artifacts are to real X-ray artifacts (unfilled rectangles and circles in the high resolution image and then downsampled).

The motivation behind adding a VAE bottleneck layer is unclear. From the description, it sounds like $\mathcal{L}_{VR}$ is the reconstruction loss between the input to BLVE and output from BLVD. This means the autoencoding is being performed with respect to the latent variables -- why does this help boost performance for the original super resolution / artifact removal task? Is it possible that you just needed to increase the number of channels to the BLU to achieve similar performance? The claim that the VAE branch was "capturing the data distribution", and that's the reason why there's some improvement is currently not very convincing.

**Detailed Comments:**

"The resultant SR image is likely contain richer information about the abnormalities" - grammar

"Auto-encoding variational bayes" - 2022? Seems like a mistake.

**Justification Of Final Rating:**

Thank you to the authors for answering some of the questions. The clarification for the artifact generation is helpful.
I maintain my original rating of weak accept. The rationale behind the variational approach still seems a bit hand-wavy, especially since it's not immediately obvious or explained very clearly why the variational formulation of the latent variables should lead to improved performance. The rebuttal argument about KL is not very convincing since that's simply a part of the VAE formulation.

**Justification Of The Preliminary Rating:**

Despite some of the questions that I had, this paper includes most of the qualities that we look for in a conference paper - clear descriptions of the method, comparisons with existing methods, ablation studies to prove the importance of each component of the model, and validations with multiple datasets.

**Questions To Address In The Rebuttal:**

Addressing the two main comments in the "weaknesses" would be helpful.

**Special Issue:**

No

---

> ### Author Response · Authors · 2024-03-16
> **Addressing the comments mentioned in the "weaknesses" and "Detailed Comments" section**
>
> $\textbf{1. Comments of the reviewer:}$
> It's unclear how close the generated artifacts are to real X-ray artifacts (unfilled rectangles and circles in the high resolution image and then downsampled).
>
> $\textbf{Author's response:}$ The original chest x-ray artifacts comprise circular shapes, various letters and digits, and lines of different types [1, 2]. We have tried to generate similar types of artifacts for our experiments. For example, the circles we have generated resemble the circular artifacts. Similarly, part of our rectangular artifacts may emulate artifacts of the shape of a line.  Furthermore, we also notice that our model can remove artifacts consisting of letters and digits which were present in the original x-ray images. We revise Section 2.4 to include this discussion.
>
> $\textbf{2. Comments of the reviewer:}$ The motivation behind adding a VAE bottleneck layer is unclear. From the description, it sounds like $L_{\text{VR}}$ is the reconstruction loss between the input to BLVE and output from BLVD. This means the autoencoding is being performed with respect to the latent variables -- why does this help boost performance for the original super resolution / artifact removal task? Is it possible that you just needed to increase the number of channels to the BLU to achieve similar performance? The claim that the VAE branch was "capturing the data distribution", and that's the reason why there's some improvement, is currently not very convincing.
>
> $\textbf{Author's response:}$ We introduce a VAE mechanism in our design. We agree that $L_{\text{VR}}$ captures the reconstruction loss. However, the loss component $L_{\text{KL}}$ helps in capturing the distribution of the data in the latent layer BLVL. The output of BLVD is generated by sampling from the distribution learnt in BLVL. Therefore, the quality of the output from BLVL and thereby the output of the proposed method is likely to be dependent on the data distribution captured in BLVL. We revise the second paragraph of Section 2.2 to include this discussion.
>
>
> $\textbf{3. Comments of the reviewer:}$ “The resultant SR image is likely contain richer information about the abnormalities.” - grammer
>
> $\textbf{Author's response:}$ We thank the reviewer for pointing out this mistake. We will rectify it in the revised version in the relevant place.
>
> $\textbf{4. Comments of the reviewer:}$ “Auto-encoding variational bayes" - 2022? Seems like a mistake.
>
> $\textbf{Author's response:}$ We thank the reviewer for pointing out this mistake. It will be rectified in the revised version.
>
> $\textbf{Reference:}$
> 1. https://www.radiologymasterclass.co.uk/gallery/chest/quality/chest-x-ray-artifact
> 2. https://radiopaedia.org/cases/brassiere-artifact

---

### Official Review · Reviewer_ArK5 · 2024-03-03

**Confidence:** 5
**Preliminary Rating:** 2
**Final Rating:** 2

**Summary:**

The authors propose an approach to address the image quality enhancement problem in Chest X-rays. In their approach, the authors build upon SR3, a diffusion model used for image super-resolution, and incorporate additional components such as bounding box and VAE loss for artifact removal. Experimental results indicate that their approach outperforms SR3 and other baseline methods in terms of performance.

**Strengths:**

1. Super-resolution and artifact removal are simultaneously performed on Chest X-rays.
2. The authors propose a learning strategy based on bounding box and VAE in addition to SR3.
3. The proposed approach outperforms the baselines in terms of PSNR and SSIM.

**Weaknesses:**

1. The mathematical foundations concerning the diffusion probabilistic models for this problem setting are missing, making the approach difficult to understand.
2. Hallucinations render this approach challenging to utilize in a clinical environment.
3. The writing style of the paper can be enhanced for better clarity and readability.

**Detailed Comments:**

1. In Figure 2, I have noticed that some changes occur in the images after using your approach, especially in the leftmost image. Have you investigated how these hallucinations affect the performance of your model from a medical point of view? Despite ESRGAN and SRGAN's inability to remove artifacts and their low performance, at least they do not alter the location and shape information of the bones. I do not think it is safe to use diffusion models for medical image super-resolution and artefact correction at the moment unless you have an approach to mitigate this issue. You may want to look at other diffusion-based methods as I2SB which is suitable also for your problem setting.
2. How do you perform the sampling of your probabilistic model? Given that your method is based on SR3, I assume it is a modified version of DDPM. Nevertheless, I see that you have changed some hyperparameters like $T$, decreasing it from 2000 to 1000. Could you describe your motivation behind such changes? Additionally, please detail your sampling methodology in Section 2.5.
3. The shift in image intensity distribution in SRGAN is noticeable. Are there any reasons why?
4. Please position the HR image at the bottom of Figure 2.
5. The purpose of the green and blue arrows in Figure 1 is unclear. Does the operation depicted in Figure 1 demonstrate how you modify an existing Residual block, or do you construct an architecture that works in a parallel fashion to U-Net?
6. Like VAEs, diffusion models are also capable of modeling the data distribution, $p(x)$. However, given the minimal improvement over the baseline method SR3, I am unsure whether it brings more merit considering the marginal computational load. Have you considered performing a statistical significance test on the methodologies?
7. Please provide mathematical descriptions of the loss terms in Section 2.3.
8. Please proofread the text for misuses of lowercase and uppercase typing, such as "super-resolution" versus "Super-resolution."
9. A discussion should be included regarding the potential limitations of your approach, as mentioned in (1).

**Justification Of Final Rating:**

Despite the authors demonstrating the significance of their method over the SR3 baseline, I remain unconvinced about how the adoption of the latent VAE aids the network in restoring the image mathematically. Furthermore, I observed that some of the visualizations are not aligned (see the third column of Figure 4), raising concerns about whether this inconsistency might impact the quality metrics. Also, upon closer examination of the visualizations, it appears that the artifacts may not completely vanish from the image but instead manifest as body parts. This raises doubts about the clinical applicability of this work, which has not been addressed in the limitations section.

**Justification Of The Preliminary Rating:**

Although the authors have introduced the BB and VAE losses on SR3 for simultaneous super-resolution and artefact removal and show some interesting results, the insufficiency in the mathematical foundations and validation of the paper weakens the strengths.

**Questions To Address In The Rebuttal:**

Mentioned in the detailed comments section.

---

> ### Author Response · Authors · 2024-03-16
> **Addressing the questions to be addressed in Rebuttal.**
>
> $\textbf{1. Comment:}$ In Figure 2, I have noticed that some changes occur in the images after using your approach, especially in the leftmost image. Have you investigated how these hallucinations affect the performance of your model from a medical point of view? Despite ESRGAN and SRGAN's inability to remove artifacts and their low performance, at least they do not alter the location and shape information of the bones. I do not think it is safe to use diffusion models for medical image super-resolution and artifact correction at the moment unless you have an approach to mitigate this issue. You may want to look at other diffusion-based methods such as I2SB which is suitable also for your problem setting.
>
> $\textbf{Response:}$ In this work, we have not investigated the quality of the output images produced by our model from a medical point of view. However, the performance measures like PSNR and SSIM used in our paper focuses on the quality of the output image based on its resemblance with the ground truth (GT) image. While PSNR focuses on the similarity between the intensities of the output image and the GT image, SSIM focuses on structural similarities between them. Our method outperforms the competing approaches (as evident from Table 1) in most occasions. This shows the ability of our method in producing output images that are similar to the GT images.  In the future, we will evaluate the quality of our output images from a medical view point. We will also look into the I2SB method. We thank the reviewer for the suggestions.
>
>
> $\textbf{2. Comment:}$ How do you perform the sampling of your probabilistic model? Given that your method is based on SR3, I assume it is a modified version of DDPM. Nevertheless, I see that you have changed some hyperparameters like T decreasing it from 2000 to 1000. Could you describe your motivation behind such changes?
>
> $\textbf{Response:}$ Through validation performance, we find that using 1000 as the timesteps yields superior results compared to using 2000 as timesteps. Thus, based on the validation performance, we decide the value of the timesteps (a hyperparameter of our model). This is already mentioned in Section 2.6.
>
> $\textbf{3. Comment:}$ The shift in image intensity distribution in SRGAN is noticeable. Are there any reasons why?
>
> $\textbf{Response:}$ The presence of the artifacts may change the intensity distributions of the input images. The SRGAN model may have more difficulty in dealing with such shifts in the distribution compared to other competing models including ours. The shift in the image intensity distribution in the output images by the SRGAN model may be the result of this difficulty that more severely affects the performance of the SRGAN model.
>
> $\textbf{5. Comment:}$ The purpose of the green and blue arrows in Figure 1 is unclear. Does the operation depicted in Figure 1 demonstrate how you modify an existing Residual block, or do you construct an architecture that works in a parallel fashion to U-Net?
>
>
> $\textbf{Response:}$ We use the SR3 baseline with additional modifications. We add the BLVE, BLVL and BLVD layers parallel to the BLU which is the bottleneck part of the U-net model. The above layers just work parallel to the bottleneck layer(BLU) of the U-net model. The Green and Blue arrows indicate only the down-sampling and up-sampling operation. We have mentioned this in the caption of Figure 1 in the revised manuscript.
>
> $\textbf{6. Comment:}$ Like VAEs, diffusion models are also capable of modeling the data distribution,p(x). However, given the minimal improvement over the baseline method SR3, I am unsure whether it brings more merit considering the marginal computational load. Have you considered performing a statistical significance test on the methodologies?
>
> $\textbf{Response:}$ To statistically evaluate the performance of our approach, we perform a statistical significance test. For each method, we perform ten rounds of experiments. At each round, we evaluate PSNR and SSIM on the VinBig and NIH dataset through. Using those values of PSNR and SSIM values for both the VinBig and NIH test data, we perform a t-test to look into the statistical significance of the difference in performance between our method and SR3. The p-values of these experiments are reported in Figure 5 of Appendix E. For NIH, the p-values for PSNR and SSIM are 0.0029 and 0.0018, respectively. For VinBig, the p-values for PSNR and SSIM are 0.0011 and 0.0158, respectively. From those results, we prove that the improvement by the proposed method is meaningful and not just by chance. Notice that all the p values are significantly less than 0.05. Therefore, we can conclude that the results of our method are statistically significantly different from those of SR3.
>
>
> Additional changes: We will position the HR image at the bottom in Figure 2, add the mathematical descriptions of the loss terms in Section 2.3, and proof-read for the mentioned misuse of lower and upper cases.

---

### Meta-Review · Area_Chair_bgpW · 2024-03-29

**Recommendation:** Accept (Poster)
**Confidence:** 4

**Metareview:**

The reviewers all appreciate the exhaustive experiments to assess the effect of the VAE that the authors added to the standard U-net (including generalization experiments that the authors ran around the clock for the rebuttal). While concerns remain about the theoretical justification of the VAE and the clinical applicability of method potentially suffering from hallucination, this is still a methodological twist that may be of interest to the MIDL community. I recommend borderline acceptance for this paper.

---

### Decision · Program_Chairs · 2024-04-05

Accept (Poster)